# All optical dynamic nanomanipulation with active colloidal tweezers

Souvik Ghosh [1] & Ambarish Ghosh[1,2]

Manipulation of colloidal objects with light is important in diverse fields. While performance of traditional optical tweezers is restricted by the diffraction-limit, recent approaches based on plasmonic tweezers allow higher trapping efficiency at lower optical powers but suffer from the disadvantage that plasmonic nanostructures are fixed in space, which limits the speed and versatility of the trapping process. As we show here, plasmonic nanodisks fabricated over dielectric microrods provide a promising approach toward optical nanomanipulation: these hybrid structures can be maneuvered by conventional optical tweezers and simultaneously generate strongly confined optical near-fields in their vicinity, functioning as near-field traps themselves for colloids as small as 40 nm. The colloidal tweezers can be used to transport nanoscale cargo even in ionic solutions at optical intensities lower than the damage threshold of living micro-organisms, and in addition, allow parallel and independently controlled manipulation of different types of colloids, including fluorescent nanodiamonds and magnetic nanoparticles.

[1] Centre for Nano Science and Engineering, Indian Institute of Science, Bangalore 560012, India. [2] Department of Physics, Indian Institute of Science, Bangalore 560012, India. Correspondence and requests for materials should be addressed to A.G. (email: ambarish@iisc.ac.in)

Controlled manipulation of nanomaterials in fluidic environment is important for both technological advances as well as to address fundamental questions in colloidal and biophysical phenomena. Among different methods of colloidal manipulation, optical tweezers[1] have been particularly successful due to their inherent versatility[2]. Conventional optical traps are built using highly focussed laser beams where the focal spot can be positioned accurately within the fluidic chamber through external optical assembly, allowing multiple[3] tweezers to carry out micromanipulation tasks in parallel, leading to many important breakthroughs in biology, materials science, and soft condensed matter physics[4]. While performance of conventional optical tweezers is limited[5] by diffraction, alternate newer strategies rely on strong near-field concentration of plasmonic nanostructures under resonant optical illumination[6]. These near-field based tweezers[7,8] can generate stronger optical potentials around them, and thereby able to trap smaller colloids[9–12] at lower illuminations intensities, compared to traditional optical tweezers. Apart from being a powerful tool for nanomanipulation, plasmonic tweezers are also important in experiments pertaining to living systems, such as to reduce the possibility of photo-damage at operating irradiation levels[13].

The strong concentration of the electromagnetic field in plasmonics confines the trapping region to the immediate vicinity of the nanostructure, typically within few tens of nanometers. This limits the speed of the trapping process by the slow diffusion of colloids to the electromagnetic hot spots in the absence of additional forces[14]. In addition, the plasmonic structures are fabricated on the surfaces of fluidic chambers[15], which do not allow dynamic manipulation of trapped colloids with the same versatility and spatial range as conventional optical tweezers operating in bulk liquids, which is a well-recognized problem[16] with this otherwise powerful technique.

The challenge that emerges is, therefore, how to combine the advantages of optical and plasmonic tweezers in a single strategy, such as to trap submicron colloids at low, biologically benign laser powers while retaining the generality of conventional optical tweezers, including dynamic, parallel, independent, and long-range manipulation in standard microfluidic chambers.

The solution demonstrated here relies in an all-optical strategy to controllably maneuver a nanotweezer for dynamic trapping and manipulation exercises, based on a "tweezer in a tweezer" configuration. The idea is schematically shown in Fig. 1, where an optical tweezer is used to trap a metallic nanodisk, which in turn functions as a plasmonic tweezer to trap colloidal objects. This technique relies on optical fields alone, through a combination of forces induced by far (optical tweezer) and near (plasmonic) field focussing. In this strategy, multiple disks can be actively and independently transported using externally controlled optical traps, where each nanodisk acts as a highly efficient active colloidal tweezer (ACT).

## Results

**Design of the tweezer.** The experimental design requires careful consideration of multiple experimental parameters, especially dimensions of the ACTs and wavelength of operation, such as to maximize the near field enhancement with minimal heating of the plasmonic structure. We consider silver disks of varying diameter and fixed thickness (50 nm) with polarized illumination traveling along the major axis of the disk. The fundamental low energy dipole mode results in strong field enhancement around the disk at the resonance wavelength, which increases with the diameter of the disk as shown in Fig. 2a. The simulation conditions are described in greater detail in Supplementary Note 1. We also calculate the steady state temperature of the surface of the disk at resonance for a fixed illumination intensity (1 mW μm$^{-2}$), considering the heat generated at the nanodisk is removed by conduction (see Supplementary Note 2) through the surrounding medium (here, water). We plot the resonance wavelength ($\lambda_{res}$) and rise of disk temperature above ambient ($\Delta T$) as a function of disk diameter in Fig. 2a. For diameter around 250 nm (corresponding to $\lambda_{res} \approx 1000$ nm), $\Delta T$ was minimal, which as discussed later (also see Supplementary Note 3) could be validated experimentally. The optimal wavelength for silver nanodisks falling in the near-infrared (NIR) regime is encouraging from the perspective of applications in bio-manipulation, considering most living cells, microorganisms and tissues have minimal absorption in the NIR[17].

The proposed strategy will work only if the plasmonic nanodisks can be trapped by an optical tweezer, i.e., if the gradient forces acting on the structures are larger than the effects due to absorption and scattering. Experiments to trap metallic nanoparticles have been attempted in the past[18–22] with focussed NIR lasers ($\lambda \sim 1000$ nm), where optical absorption is lower. Particularly relevant is the work[20] by Bosnanc et al., where silver nanoparticles of different radii (20–275 nm) were trapped using focused laser illumination at 1064 nm with varying optical power ranging between 50 and 300 mW. The high optical power produced a large trapping potential, which was necessary to counter the strong position fluctuations of the nanoparticles due to Brownian motion. The corresponding illumination intensity is significantly larger than what is typically used with plasmonic tweezers and defeats the advantage of using a plasmonic tweezer, i.e., trapping colloids with low optical powers.

The optical intensity to trap the ACTs must be reduced while ensuring there is enough optical field around each ACT to function as an efficient plasmonic trap. In the present approach, we add a dielectric microrod on each metal nanodisk to reduce the Brownian diffusivity, which keeps the optical properties (e.g., $\lambda_{res}$) essentially unchanged while reducing the optical power required to trap the ACTs by a significant amount. Schematic of this hybrid structure of the ACT is shown in Fig. 2b, while diffusivities of rods of diameter 250 nm and different lengths are

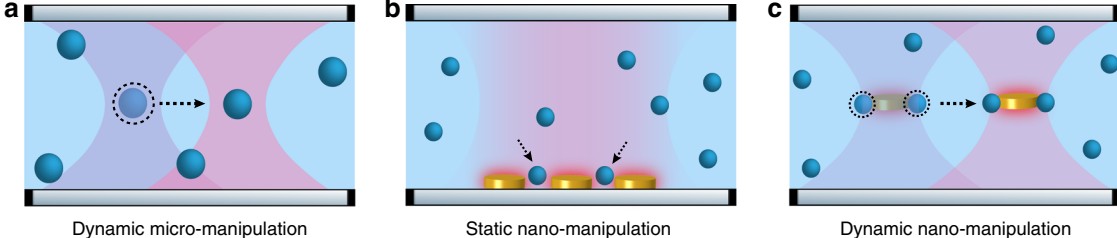

**Fig. 1** Schematic illustration of the concept of ACT. **a** Dynamic manipulation of micro-objects with conventional optical tweezers. **b** Trapping of nano-objects with plasmonics based nanotweezers at predefined locations. **c** Dynamic nanomanipulation with ACT by combining far- and near-field focusing of light

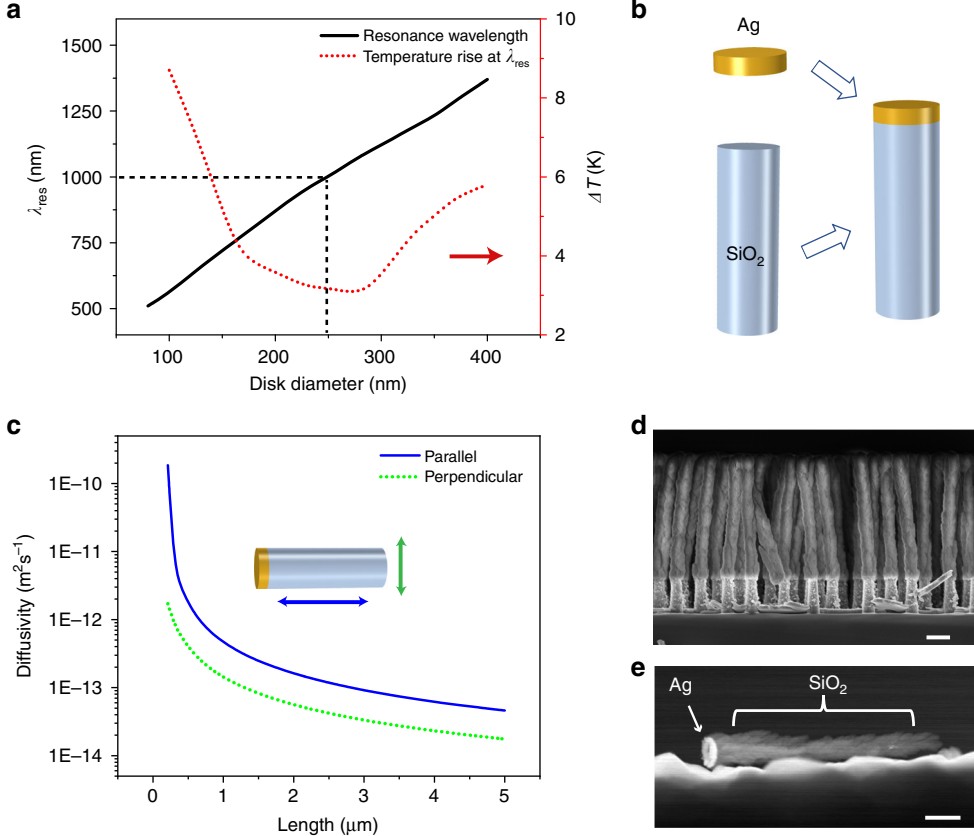

**Fig. 2** Active colloidal tweezers. **a** Resonance wavelength and temperature rise of the nanodisk immersed in water plotted as a function of disk diameter. **b** Design of an ACT formed by integration of the plasmonic nanodisk with a dielectric nanorod. **c** Brownian diffusivities[55] of nanorods as a function of length, along parallel and perpendicular directions to the long axis. **d** Scanning electron microscope (SEM) image of an array of ACTs on the substrate. Scale bar is 500 nm. **e** Transmission electron microscope (TEM) image of the single ACT released from the substrate. Scale bar is 300 nm

shown in Fig. 2c. The Brownian fluctuations reduce by ~1000 when a rod of 2.5 μm is used compared to a disk of diameter 250 nm and thickness 50 nm. As we show later, these hybrid structures can be trapped tightly with optical intensities as low as 0.8 mW μm$^{-2}$, and at the same time, generate strong near fields to act like plasmonic tweezers.

**Demonstration of trapping and manipulation**. We discuss the fabrication of ACTs in detail in the methods section and only mention the main steps here. We combined colloidal lithography technique and reactive ion etching (RIE) to make an array of nanopillars of diameter 250 ± 15 nm (see Supplementary Fig. 3a) on a silicon wafer. This was followed by deposition of 50 nm of silver to make the plasmonic nanodisk and oblique angle deposition[23] of silica micropillar of length 2.5 μm. Note all the fabrication techniques are scalable yielding ~10$^8$ ACTs per cm$^2$ of the wafer (see SEM image in Fig. 2d). To release the individual ACTs, the substrate was sonicated in a fluid. As seen from the TEM image in Fig. 2e, the ACTs break off the wafer while keeping the silver and silica parts intact.

The experiments were performed with an elliptically polarized laser source (λ = 1064 nm) coupled to an inverted optical microscope using a 100×, 1.4 NA oil immersion objective (see Supplementary Fig. 4). All experiments were carried out inside a standard microfluidic chamber made of glass, which contained a suspension of ACTs as well as cargo in the form of colloidal particles of different sizes and compositions. The ACTs could be maneuvered inside the microfluidic chamber by either moving the position of the laser focus

through external optical assemblies, or moving the microscope stage and therefore the chamber, with respect to the ACTs trapped at the laser focus.

We have tested the trapping capabilities of the ACTs with different materials and in different fluids. In Fig. 3b, we show trapping and release of 100 nm fluorescent nanodiamonds (FND biotech) by an ACT. The ACT was held static within an optical trap as shown in the schematic Fig. 3a (see Supplementary Movie 1). These nanodiamonds hold great promise in quantum technologies, so their controlled manipulation can open many new and exciting possibilities[24–27].

Ideally, a nanotweezer should be applicable in different types of fluids—especially common biological fluids which are ionic in nature[28]. As a proof of concept, we demonstrate trapping and subsequent transport and release of a 200 nm (Bangs Laboratories, Inc.) fluorescent polystyrene (PS) particle in phosphate buffer saline as shown in sequence of images in Fig. 3c (see Supplementary Movie 2). Clearly the ACTs addresses some of the major limitations of current state-of-the-art in optical manipulation: (i) Their trapping performance is not limited by the diffraction limited focussing of light, allowing trapping smaller colloids at lower laser powers. (ii) They can be actively maneuvered and therefore allow selective cargo transport. (iii) ACTs can be operated in standard microfluidic chambers, implying specialized nanophotonic substrates are not necessary. (iv) Due to its all-optical nature, ACTs are applicable in different classes of media, including iconic solutions, where many of the alternate nanomanipulation strategies may not be suitable[29–31].

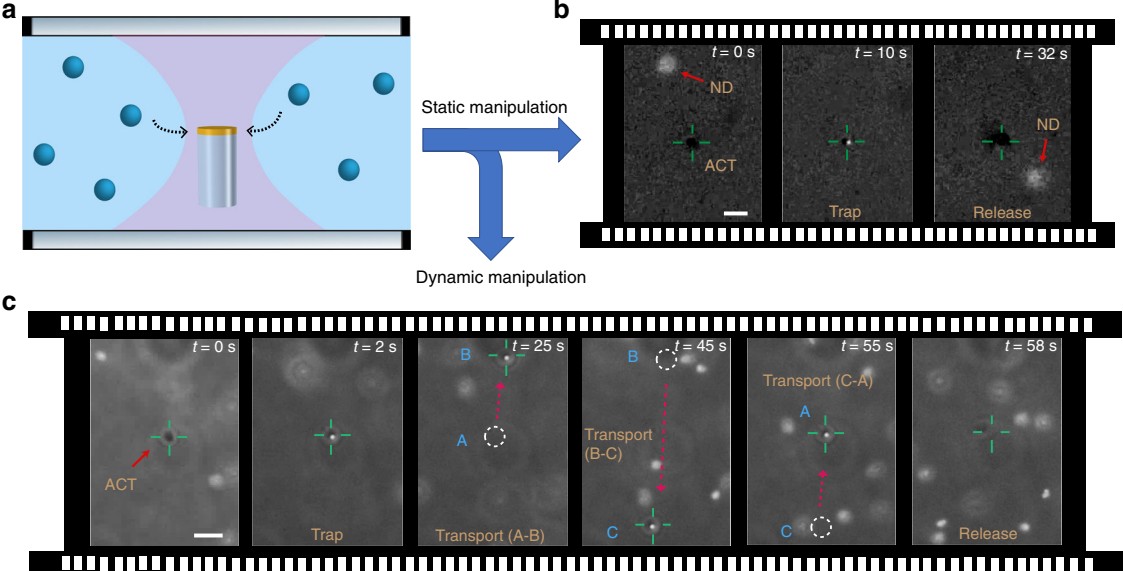

**Fig. 3** Demonstration of static and dynamic manipulation by ACTs. **a** Schematic of an ACT held in liquid with a focused laser beam. **b** Trapping of 100 nm fluorescent nanodiamonds with illumination intensity 4 mW µm$^{-2}$ and its subsequent release by turning off the illumination. **c** Trapping, transport and release of 200 nm fluorescent polystyrene bead in PBS buffer. All scale bars correspond to 3 µm

**Trapping configurations**. The optical property of an ACT was dominated by the highly polarizable silver nanodisk, which was trapped at the focal spot. For tight focussing (spot size ~1 µm), the ACT was oriented along the beam propagation direction (i.e., $z$-axis in Fig. 4a), with the dielectric rod always remaining under the focus. This occurred because the silver nanodisk was subject to higher radiation pressure compared to the silica part, which pushed the metal part higher as the ACT got trapped in the vertical orientation. When the laser spot size was increased (~10 µm) through further defocussing, a wide shallow trap was formed, which is referred to as optical bowl in the literature[32–34]. This was achieved by focusing the laser beam at a plane higher (along beam propagation direction, i.e., $z$-axis) than the plane of observation inside a microfluidic chamber, resulting in a weak two-dimensional trap. The beads experience radiation pressure upward which is balanced by the electrostatic repulsion due to the top surface of the fluidic chamber. The $z$-position of the focal plane was controlled by tuning the divergence of the beam entering the microscope objective. Here, the ACT achieved a horizontal orientation (see Fig. 4a, horizontal) loosely trapped within the illuminated region.

In Fig. 4b, we show the calculated extinction spectrum of the nanodisks as a function of wavelength, where the illumination wavelength was chosen to be slightly detuned (red-shifted) to the resonant modes where the positive sign of the real component of polarizability leads to an attractive gradient force[35]. The electric field enhancement for the vertical configuration at 1064 nm extends all around the plasmonic disk in the $x-y$ plane (see Fig. 4c). The minor difference between points I and II occurred because the light was taken to be elliptically polarized with $I_{max}:I_{min}\sim9:4$, same as the experimental illumination. As seen in Supplementary Movie 3, intensity as low as 0.8 mW µm$^{-2}$ was sufficient to generate enough enhanced local intensity to trap 400 nm PS particles. The distribution of the trapped particles (see Fig. 4e) was symmetric around the ACT, which further validates the calculated field distribution.

On the other hand, the spatial distribution of the trapped particles was found to be asymmetric in the horizontal configuration (see Fig. 4f and Supplementary Movie 4), showing zero trapping along the direction of light propagation. This was a direct validation of the field distribution calculated in the $y-z$ plane (see Fig. 4d) where no field localization was observed in point IV as opposed to strongly localized fields at point III.

An interesting advantage of the horizontal configuration is the possibility of having multiple ACTs, which can be used to manipulate a collection of colloidal beads with a single beam (see Supplementary Movie 5). On the other hand, vertical configuration is advantageous for manipulation with higher spatial control and resolution to move and collect target colloids. As expected, the colloids were trapped more tightly in the vertical configuration, which originated due to the reduced position fluctuations of the ACTs (measurements shown in see Supplementary Fig. 5).

**Measurement of trap parameters**. An important parameter for any optical tweezer is the minimum laser intensity to trap particles (here, colloidal beads) of different sizes. This is shown in Fig. 5a. The minimum trapping intensity in vertical configuration with a strongly focused laser was constant for PS beads of different sizes. This was because minimum optical power required to trap ACTs, here 0.9 mW µm$^{-2}$ was higher than the threshold intensity required for plasmonic trapping of the colloidal beads. The size dependence was visible for the horizontal configuration, where the ACTs could be trapped at lower illumination intensities.

In Fig. 5b, we plot the estimated plasmonic trapping potential for a 100 nm PS bead at different distances from the ACT, located at the positions marked in Fig. 4c, d. The calculations confirm the trapping potential at position IV to be less than $k_BT$ even at 10 mW µm$^{-2}$, as evident from absence of trapped beads in these positions (see Fig. 4f). The potential for all other configurations were significantly deeper, ranging between 8 and 12 $k_BT$. Note, lower (~4 $k_BT$) potential depths have been used to trap particles through plasmonic field confinement in the past[9,36–40]. We have also checked the variation of trapping force due to the distribution of sizes of nanodisks to be less than 15% (see Supplementary Fig. 3b).

Figure 5c shows the histogram of the displacements of a 40 nm particle while it was trapped by a static ACT (laid on the bottom surface of the chamber). The corresponding Gaussian fits were

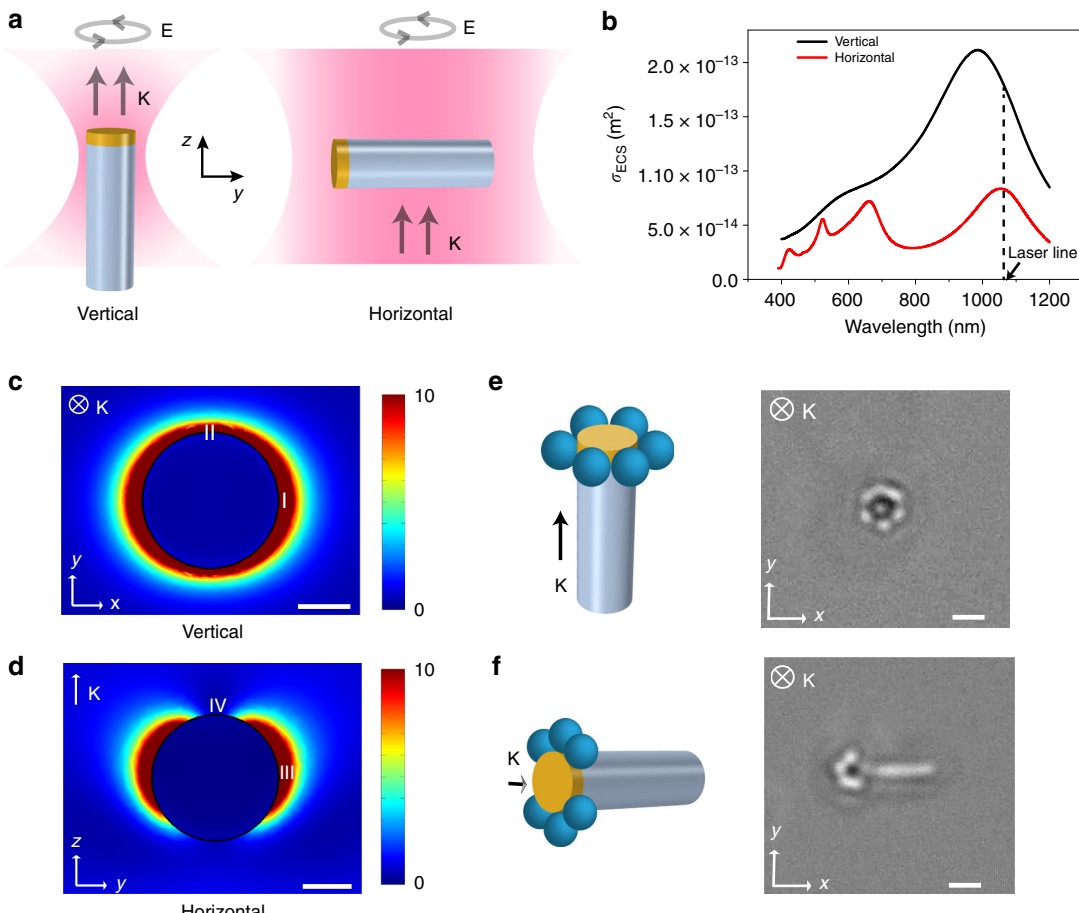

**Fig. 4** Trapping configurations. **a** Different orientations of ACT for elliptically polarized laser beam propagating along z-axis. With a tightly focused laser spot, the ACT aligns with the propagation direction resulting in a vertical configuration. Increasing the spot size causes the ACT to prefer a horizontal configuration while remaining trapped loosely within the illuminated region. **b** Extinction cross section vs wavelength for 250 nm silver nanodisks for vertical and horizontal configuration. **c**, **d** Enhancement of electric field for the nanodisks in vertical configuration and horizontal configuration under 1064 nm illumination. Both scale bars are 100 nm. **e**, **f** Trapping of 400 nm PS particles in plasmonic near-field for vertical and horizontal configuration. Both Scale bars are 1 μm

used to calculate the variance of the position fluctuation under confinement which was used to measure the trap stiffness ($\kappa$) of the ACT as well as the optical trap that holds the ACT using $\frac{1}{2}\kappa x^2 = \frac{1}{2}k_BT$. The trap stiffness due to static ACT normalized to incident power, $\kappa_p = \frac{\kappa}{P}$, has been compared with standard Gaussian beam optical traps and different plasmonic trapping strategies in Table 1. We have estimated the stiffness of the trap due to static ACT to be 0.245 fN nm⁻¹ mW⁻¹, which is comparable to other plasmonic tweezers and about 200 times higher than conventional optical tweezers when scaled to particles of 40 nm (assuming a cubic dependence on the particle size). Also relevant are the fluctuations of an ACT in vertical configuration which is shown in Fig. 5d. The well stiffness of the focused optical trap was found out to be 1.78 fN nm⁻¹ mW⁻¹, which resulted in strong confinement of the ACT within the optical trap. The presence of two trapping stiffnesses is a unique manifestation of the "tweezer in a tweezer" configuration presented here.

In Fig. 5e, we highlight the position of the trapped beads within the focal volume. Without the ACT, a particle is trapped at the laser focus where the trapped object is located at the region of highest intensity. However, in the present scheme, ACTs occupy the position of maximum intensity implying the effective optical power incident on the trapped colloids is a fraction of applied power. For a 400 nm PS bead, this correspond to a further

reduction of effective incident laser power by a factor of ~10, which can provide crucial advantages in biomanipulation applications[41]. Also, relevant is the temperature rise in the system due to absorption of incident light. Using temperature dependence of fluorescence of the dye molecules in the trapping region, we have measured (see Supplementary Note 3) the temperature rise to be ~13K at an intensity level 10 mW μm⁻², which corresponds to a trapping potential larger than $10k_BT$ for a 100 nm particle. Note this is lower than the optical damage threshold of a living bacteria [41] implying interesting possibilities in biomanipulation experiments. Please note that, the measurements provided in Fig. 5a correspond to applied laser intensities, and not effective (reduced) intensities, as discussed above.

**Manipulation capabilities of ACTs.** Next, we check the trapping performance of the ACTs with very small colloids. As shown in Supplementary Movie 6, we could observe trapping for 40 nm (see inset of Fig. 6a) fluorescent PS nanoparticles (Bangs Laboratories, Inc.) by ACTs held in vertical configuration. In Fig. 6a, we show the trapping and release of individual colloids (see Supplementary Movie 7), as seen in the temporal variation of the fluorescence intensity. For this experiment, we use an ACT attached to the chamber surface (implying trapping in horizontal configuration), such as to reduce difficulties in imaging

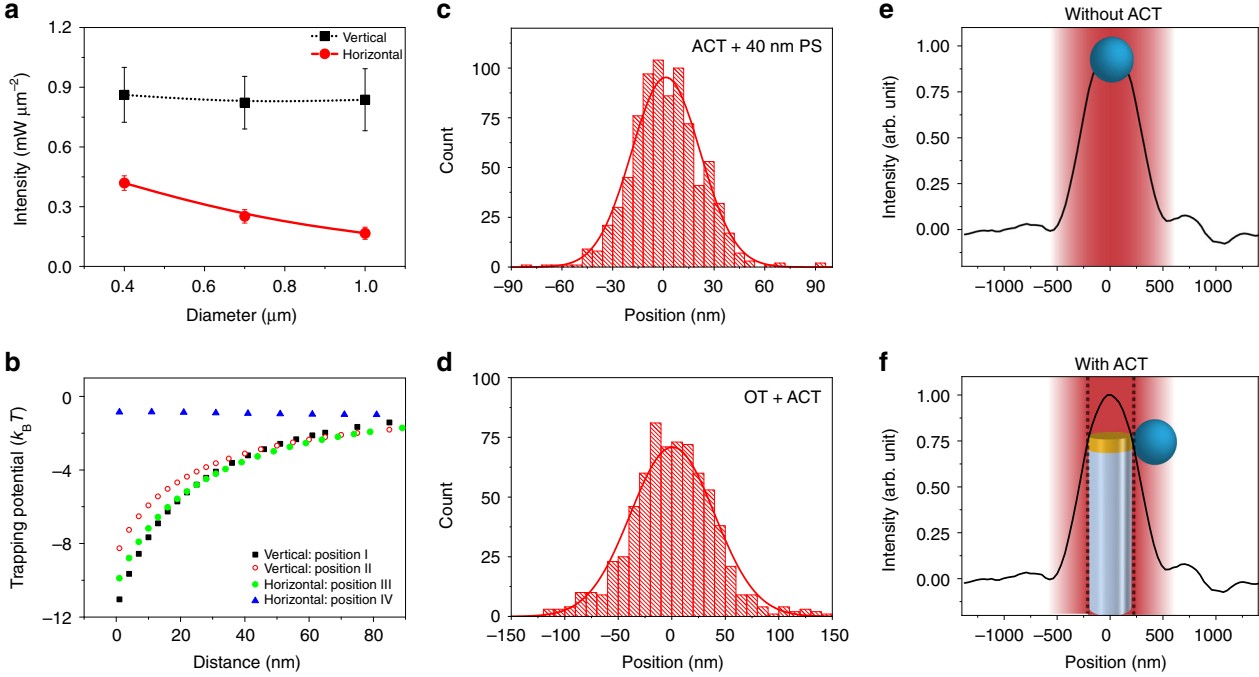

**Fig. 5** Analysis of ACT performance. **a** Minimum illumination intensity to trap beads (Spherotech Inc.) as a function of the bead size for two configurations of the ACT. **b** Calculated plasmonic trapping potential for a 100 nm PS particle as a function of distance from the nanodisk. The simulations were performed in the two configurations, for positions of the PS particles as marked in Fig. 4c, d. **c** Histogram of position fluctuations of a 40 nm PS particle trapped on a static ACT. **d** Histogram of position fluctuations of a trapped ACT in an optical tweezer (OT) in vertical configuration. **e**, **f** Measured intensity profile of the focused gaussian beam shown in black lines. Schematic shows position of the trapped bead without and with the ACT within the optical trap: peak illumination intensity is blocked the ACT, implying trapped colloids are in a relatively lower intensity region of the Gaussian beam

| Table 1 Comparison of trap stiffness for different optical traps scaled for a 40 nm PS sphere | | | | | |
|---|---|---|---|---|---|
| **Trapping platform** | **Strategy** | **Maneuverability** | **Particle radius ($r$) (nm)** | **Stiffness ($\kappa_p$) (fN nm$^{-1}$ mW$^{-1}$)** | **Scaled stiffness (×20 nm/$r$)$^3$ (fN nm$^{-1}$ mW$^{-1}$)** |
| Focused Gaussian beam[18,54] | Optical | Yes | 110 | 0.2 | $1.2 \times 10^{-3}$ |
|  |  |  | 19 | $0.7 \times 10^{-3}$ | $0.8 \times 10^{-3}$ |
| Nanoaperture[37] | Plasmonic + self-induced back action | No | 25 | 6.6 | 3.38 |
| Nanodisk[12] | Plasmonic | No | 55 | 7.6 | 0.36 |
| Double nanoblock[38] | Plasmonic | No | 50 | 4.6 | 0.29 |
| ACT (this work) | Plasmonic + Optical | Yes | 20 | 0.24 | 0.24 |
| *SIBA* self-induced back action | | | | | |

due to thermal fluctuations. It is also possible to manipulate collection of 40 nm particles, as shown in Fig. 6b, where the ACT was trapped in the vertical configuration and maneuvered across the chamber.

In Fig. 6c, we demonstrate parallel and independent manipulation of 300 nm fluorescent magnetic particles (Spherotech) using two ACTs. Such magnetic beads are routinely used in various microfluidic sensing technologies[42]. We used a fast-scanning galvo-mirror to trap two ACTs simultaneously. Initially, each ACT was used to trap one magnetic particle. Subsequently, the tweezers were independently maneuvered to two different locations that was nearly 20 μm apart, where the particles were released sequentially by turning the illumination off. As far as we know, similar independent and parallel control has not been achieved with any other near-field trapping scheme. Note, to make similar manipulations with conventional optical tweezers will require an order of magnitude higher illumination intensity than what was used in the present experiments.

## Discussion

We envision several interesting possibilities to enhance the future capabilities of ACTs: (i) Holographic tweezers[43] can allow massively parallel, yet independent manipulation of trapped colloids by large number of ACTs. (ii) The choice of illumination wavelength in the present experiments was motivated by the ubiquitous use of 1064 nm in conventional optical traps; accordingly, we envision a seamless integration of ACTs to existing tweezer systems. (iii) Although we used the same laser beam to trap the ACTs and generate resonantly induced electric fields localized around the plasmonic disks, future designs can incorporate different wavelengths for trapping and generation of near fields. (iv) Optimal design of plasmonic nanostructures allows new capabilities, e.g., bow-tie[44] or other strongly resonant nanoantennannes[40] for enhanced field confinement, using self-induced back action[45] (SIBA) forces for higher trapping efficiency, enantioselective manipulation for structure-specific nanomanipulation[46], etc. (v) Similarly, one may use higher order laser modes, e.g.,

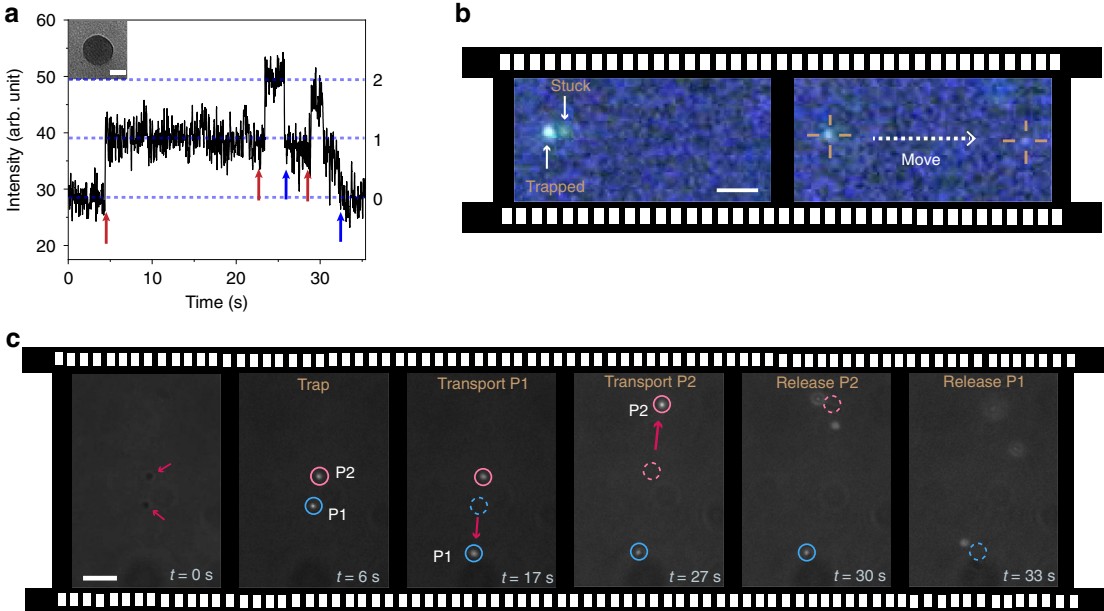

**Fig. 6** Manipulation capabilities of ACTs. **a** Trapping and releasing of 40 nm (TEM image is shown in inset, scale bar 20 nm) fluorescent PS particles by ACT (illumination intensity 32 mW μm$^{-2}$). The red and blue arrows indicate individual nanoparticles enter and exit the trap respectively. See Supplementary Movie 7. **b** Manipulation of a collection of few 40 nm PS particles by moving the stage (scale bar 1 μm). **c** Demonstration of parallel and independent control in nanomanipulation by two ACTs used for independently trapping, transporting and releasing (see Supplementary Movie 8) 300 nm fluorescent magnetic particles (scale bar 3 μm)

Laguerre–Gaussian and Bessel beams to engineer the far-field, e.g., for dynamical manipulation of the ACTs within the trap for new manipulation schemes[47–49].

In conclusion, we have described a unique "tweezer in a tweezer" configuration based on a synergy between far- and near-field optical focussing. As far as we know, this is the first all-optical technique for dynamic nanomanipulation in the bulk. Various alternate approaches have relied on combining plasmonic tweezers with additional forcing schemes, such as direct mechanical manipulation of tethered nano-patterned optical fibers[50], integration with magnetic nanorobots[51], electro-kinetic flows[14,52], thermal forces in conjunction with specific chemical moieties[29–31,53], etc. The ACTs discussed here could be operated remotely to trap and maneuver subwavelength colloidal cargo, using lower optical power than conventional gaussian beam tweezers[5,54]. They are mass-produced and thereafter can be integrated seamlessly to standard lab-on-chip devices as well as existing optical tweezer systems, such as to carry out colloidal manipulation of micron and submicron cargos. This technology may enable isolation, manipulation, and chip-level assembly of nanomaterials such as nanocrystals, fluorescent nanodiamonds and quantum dots, and allow noninvasive manipulation of fragile bio-specimens, such as bacteria, virus, and various macromolecules.

## Methods

**Fabrication**. The fabrication of ACTs is schematically described step-by-step in Supplementary Fig. 1. We used a Langmuir–Blodgett trough to deposit a monolayer of colloidal beads (made of PS) on a substrate (here Si wafer) using the Langmuir–Blodgett method, which is depicted as circles in Supplementary Fig. 1. The diameter of the beads was reduced from 400 nm to about 250 nm by etching with oxygen plasma (see SEM image, Supplementary Fig. 2a). We then deposit 200 nm coating of Chromium on individual beads using glancing angle deposition, where the substrate was kept at an extreme angle with respect to the source of evaporated material (see SEM image, Supplementary Fig. 2b). The evaporated Cr covers the upper half of the spherical bead, thereby acts as a mask (protection layer) for the next step. The entire substrate was then etched vertically by RIE using Cl-gas to a depth greater than 500 nm, leaving the Cr-coated bead above a Si pillar of 250 nm diameter. The Cr-coated beads were removed by sonication in acetone

leaving the flattop Si pillars of 250 nm diameter (see SEM image, Supplementary Fig. 2c). This was followed by vertical deposition of 50 nm Ag over the pillars, resulting in plasmonic disks of 250 nm diameter and 50 nm thickness. To fabricate the dielectric part, SiO$_2$ thin film was grown again using glancing angle deposition. Here, the substrate was kept at extreme angle (~86°) with respect to the evaporation source and rotated quickly (1.5 rpm for deposition rate ~0.5 nm s$^{-1}$), resulting in the formation of silica nanorods (see SEM image, Supplementary Fig. 2d). For better adhesion of SiO$_2$ with Ag nanodisk, a 3 nm Ti layer was deposited as adhesion promoter. This fabrication method is wafer scale with a yield of about 10$^8$ ACT nanostructures per cm$^2$.

## Data availability

The data that support the findings of this work are available from the corresponding author upon reasonable request.

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

## Acknowledgements

We thank Prof. Deepak Saini, Debayan Dasgupta, Gouri Patil, Praneet Prakash, Reshma V.R. and V. Balaswamy for helpful discussions, DST-BRICS and DBT for funding. We thank MHRD, MeitY, and DST for supporting the facilities at CeNSE.

## Author contributions

S.G. and A.G. conceived and developed the idea. S.G. built the optical setup and the numerical model and performed the experiments and numerical simulations. Both authors brainstormed and discussed extensively during this research and prepared the manuscript together. A.G. supervised the work.

## Additional information

**Competing interests:** The authors declare no competing interests.

**Peer Review Information** *Nature Communications* thanks Reuven Gordon, Euan McLeod and other, anonymous, reviewers for their contribution to the peer review of this work. Peer reviewer reports are available.

