## [Peer Review File · Nature Communications]

Reviewers' comments:

Reviewer #1 (Remarks to the Author):

The manuscript entitled, "All optical dynamic manipulation with active colloidal tweezers," presents a specially designed plasmonic tool for optically trapping and manipulating nanoparticles. This tool is itself optically trapped using a conventional focused Gaussian beam optical trap. The tool consists of a silver nanodisc at the end of a micro-scale cylindrical silica rod. The authors demonstrate that they can trap a range of different particles, with the smallest being 40 nm fluorescent polystyrene particles. Simulations of the trapping forces and diffusivity are included.

Because of the novelty of the plasmonic tool and the experimental demonstration of its use, I think the manuscript could ultimately be suitable for Nature Communications. However, in its current form, it is lacking a rigorous, quantitative comparison to the performance of conventional optical tweezers, which I feel is necessary for a paper at the level of Nature Communications. In the context of this manuscript, relevant performance metrics could perhaps be: (1) the smallest particle that can be optically trapped, (2) the minimum irradiance at the particle location necessary to trap it, or (3) the maximum optical potential well depth that can be achieved before reaching a damage threshold of a given material. It may not be necessary for the authors to address all of these aspects, but some quantitative comparison to standard Gaussian beam traps should be given.

(1) In terms of particle size, smaller colloids have been trapped in conventional optical tweezers. For example, 18 nm diameter gold nanoparticles are trapped in:

Hansen, P. M.; Bhatia, V. K.; Harrit, N.; Oddershede, L. Expanding the Optical Trapping Range of Gold Nanoparticles. *Nano Lett.* 2005, 5, 1937-1942.

and 9.5 nm diameter gold nanoparticles are trapped in:

Hajizadeh, F.; Reihani, S. N. S. Optimized Optical Trapping of Gold Nanoparticles. *Opt. Express* 2010, 18, 551-559.

From my rough calculations, the forces on gold nanoparticles are approximately 10 times stronger than those on polystyrene nanoparticles at this wavelength, but the force scales with particle volume. So the trapping of 9.5 nm diameter gold particles would be approximately as difficult as the trapping of 20 nm diameter polystyrene particles.

If the authors want to claim improved ability to trap the smallest possible particles, then they should quantitatively show how much smaller of a particle they can trap compared to a conventional focused Gaussian beam.

(2) With regard to minimum irradiance needed on a particle to trap it, there is again no quantitative comparison to a standard Gaussian beam trap. On lines 199-200, the authors state that the irradiance due to the using the ACT is about 10 times lower due to the ACT, making the irradiance at the particle on the order of 0.09 mW/ μm^2 . However it is not clear if this only takes into account the off-axis position of the particle relative to the incident beam, or whether it also takes into account the $\sim 10\times$ plasmonic increase in electric field intensity due to the near-fields of the ACT as shown in Fig. 4c. Also, how do these values compare to what is required using a conventional Gaussian beam trap?

(3) Perhaps a deeper potential well depth for nanoparticles could be achieved using the ACT without material damage than is possible with a standard Gaussian beam trap? A recent reference (from our own group) that investigates the maximum potential well depth / maximum force for conventional traps of nanoparticles is:

Melzer, J. E.; McLeod, E. Fundamental Limits of Optical Tweezer Nanoparticle Manipulation Speeds. ACS Nano 2018, 12, 2440-2447.

Some other more minor issues that should be addressed prior to publication are:

(a) In lines 13, 128, and 136, I would recommend using the term "ionic solution" rather than "ionic fluid". The terms "ionic fluid" and "ionic liquid" typically refer to substances that are purely composed of ions and are not water-based.

(b) More detail should be given on how the Brownian diffusivities are calculated in Fig. 2c. Currently only a reference to a textbook is given. Do the calculations assume 2D or 3D diffusion? Is the rod modeled as a spherical particle, or is it truly modeled as a rod? If the particle is spherical, what is the equivalent diameter based on: is it particle volume, particle length, or particle width? If the rod is truly modeled as a rod, how is the diffusivity calculated, and is the potential for tumbling taken into account?

(c) In the supplemental info and in the Fig. 4 caption, the plasmonic tools appear to be called CPTs rather than ACTs. The acronym should be consistent.

(d) On line 152, the authors write that the laser spot size was increased to 10 μm "through further defocusing". It is not clear to me what is happening here. I would have thought that defocusing of the objective would primarily move the trap along the z-axis, but not significantly change the spot size. Is this a 2D trap and not a 3D trap? Or is it a different lens in the optical system that is being defocused?

(e) On line 156, the authors write, "the illumination wavelength was found to be slightly red-shifted to the resonant modes." I do not understand this. I thought the illumination wavelength was fixed at 1064 nm. Is there some nonlinear process that is changing the illumination wavelength inside the system?

(f) Where were all of the different nanoparticles acquired? Were they bought from a company or synthesized?

Sincerely,
Euan McLeod
College of Optical Sciences
University of Arizona

Reviewer #2 (Remarks to the Author):

Dear Editor,

This paper reports the dynamic manipulation of particles by employing an ACTS. The authors successfully fabricated metallic nanodisks of varying diameter and fixed thickness on dielectric nanorod. They, further, trapped 100 nm fluorescent nanodiamonds with intensity 4 $\text{mW}/\mu\text{m}^2$, beads in PBS solution. However, major part of results in this paper were done by COMSOL, and experimental part was very poor and lack of experimental analysis, such as trapping force, stiffness, quality factor and dynamic motions of the trapping system of ACT itself, ACT with NV diamonds and ACT with 200 nm polystyrene. All these ACT, NV diamond, and 200 nm polystyrene can be easily trapped separately at different and much lower intensity levels with other systems. Their trapping of group particle seems to be not so much different in term of physics with a group of particles of clusters from Toussaint and Bryan in a Nano Letter sometimes back. I am not sure this is enough to justify publication in Nan. Com; especially considering there is a priori nothing

novel about plasmon-assisted trapping and this configuration does not perform particularly better than prior ones. The reviewer feels that the scope of this manuscript may be more suited to another journal after lots of corrections of course, and for sure it is not for a Nature Com.

Some comments:

- 1) Figure 5 c, I don't know what they want to show in this figure, it looks not like a near-field, the laser beam was not scattered and so on,.. with the presence of object. It is meaning less that figure. Again similar problems with the figure 6a and it is hard to convince people that this is a trap for few seconds or it was just the flow of water that caused by the heating and this may lead the particle flowing around the ACT and got stuck for few second and release.
- 2) There is also a lack of discussion of where the trapped particle is in three dimensions, which is important for a broadly read journal such as Nat. Com., rather than those who are experts in nanoplasmonic structures.

Reviewer #3 (Remarks to the Author):

In the manuscript: "All optical dynamic nanomanipulation with active colloidal tweezers," the authors present a hierarchical scheme for manipulating small objects with plasmonic tweezers attached to nanorods. This is an original approach to achieve such manipulation that will be of great interest to the community and I believe that it may be suitable for Nature Communications. I do have some comments that should be addressed prior to publication.

1) My greatest concern is that the inclusion of an isolated plasmonic structure will result in substantial heating at even moderate powers. This is basically the premise of thermoplasmonics. The authors say that the heating was minimal and they provide a plot to show only a few degrees temperature increase; whereas other works have shown hundreds of degrees temperature increase. Is there any experimental evidence (e.g. changes in fluorescence, changes in Brownian motion of trapped particle) to support their claim?

2) The authors are quite careful to say that the present scheme allows for lower illumination intensities (note typo "illuminations intensities"). They are consistent in this claim saying that the overall power is lower to trap. This is true and certainly a good feature; however a common error in plasmonic trapping is to say that the plasmonic particle creates greater forces that are required to overcome diffusion of smaller particles. The authors make similar claims in the same sentence referring to the larger gradient forces. "These near-field based tweezers can generate stronger gradient forces and thereby able to trap smaller colloids". It is not the strength of the gradient force that allows for trapping smaller particles -- it is the local intensity. The potential well created by that intensity has a depth defined by the highest intensity region. It does not matter how localized or spread out that intensity is -- the depth of the well is defined by that intensity, and so the greater forces by plasmonic tweezers do not allow trapping smaller particles. They give greater localization of the trap, and trapping with lower overall power, but a conventional tweezer system will trap particles of the same size as a plasmonic one if the local intensity at the maximum is the same. Making this point clear in the manuscript is suggested, since it is misleading at best (and incorrect at worst) the way it is presented. (Again later in the manuscript they say "... sufficient to generate enough gradient force to trap..." -- also giving the wrong impression that greater gradient forces allow trapping smaller particles, when energy-based considerations show that smaller forces over a larger distance would be just as good. A general readthrough of the manuscript to carefully eliminate these types of claims is suggested.)

Reviewers' comments:

Reviewer #1 (Remarks to the Author):

The manuscript entitled, "All optical dynamic manipulation with active colloidal tweezers," presents a specially designed plasmonic tool for optically trapping and manipulating nanoparticles. This tool is itself optically trapped using a conventional focused Gaussian beam optical trap. The tool consists of a sliver nanodisc at the end of a micro-scale cylindrical silica rod. The authors demonstrate that they can trap a range of different particles, with the smallest being 40 nm fluorescent polystyrene particles. Simulations of the trapping forces and diffusivity are included. Because of the novelty of the plasmonic tool and the experimental demonstration of its use, I think the manuscript could ultimately be suitable for Nature Communications. However, in its current form, it is lacking a rigorous, quantitative comparison to the performance of conventional optical tweezers, which I feel is necessary for a paper at the level of Nature Communications. In the context of this manuscript, relevant performance metrics could perhaps be: (1) the smallest particle that can be optically trapped, (2) the minimum irradiance at the particle location necessary to trap it, or (3) the maximum optical potential well depth that can be achieved before reaching a damage threshold of a given material. It may not be necessary for the authors to address all of these aspects, but some quantitative comparison to standard Gaussian beam traps should be given.

We thank the reviewer for his constructive comments and suggestions. We have incorporated the suggested changes in the revised version, and this has resulted in a significantly improved manuscript.

Stiffness is a suitable parameter to estimate and compare different optical traps. We have estimated the trap stiffness (κ) of ACT by correlating it with the position fluctuations, using $\frac{1}{2}\kappa\langle x^2 \rangle = \frac{1}{2}k_B T$. Fig. 5d presents histograms of displacements for a 40 nm dielectric particle trapped on an ACT and the corresponding gaussian fits that were used to measure the variance of the particle displacements. The measured variance corresponds to a normalised (to power) stiffness $\kappa_p = \frac{\kappa}{p} = 0.245 \text{ fNm}^{-1}\text{mW}^{-1}$ which is about 200 times higher than standard Gaussian beam traps. We have given a quantitative comparison of ACT trapping stiffness with conventional optical tweezer as well as other plasmonic tweezers in Table 1. The table also contains stiffness scaled to the volume of the trapped particle, as would be the case for particles significantly smaller than the wavelength. The table shows how the ACT allows maneuverability like conventional optical tweezers, but with significantly higher stiffness.

The following has been added in main text:

“Figure 5c shows the histogram of the displacements of a 40 nm particle while it was trapped by a static ACT (laid on the bottom surface of the chamber). The corresponding Gaussian fits were used to calculate the variance of the position fluctuation under confinement which was used to measure the trap stiffness (κ) of the ACT as well as the optical trap that holds the ACT using $\frac{1}{2}\kappa\langle x^2 \rangle = \frac{1}{2}k_B T$. As shown in Fig. 5c, trap stiffness due to static ACT normalised to incident power, $\kappa_p = \frac{\kappa}{p}$, has been compared with standard Gaussian beam optical traps and different plasmonic trapping strategies in Table 1. We have estimated the stiffness of the trap due to static ACT to be $0.245 \text{ fNm}^{-1}\text{mW}^{-1}$ which is comparable to other plasmonic tweezers and about 200 times higher than conventional optical tweezers when scaled to particles of 40 nm (assuming a cubic dependence on the particle size). Also relevant are the fluctuations of an ACT in vertical configuration. As shown in Fig. 5d, the well stiffness of the focused optical trap was found out to be $1.78 \text{ fNm}^{-1}\text{mW}^{-1}$, which resulted in strong confinement of the ACT

within the optical trap. The presence of two trapping stiffnesses is a unique manifestation of the “tweezer within a tweezer” configuration presented here.”

(1) In terms of particle size, smaller colloids have been trapped in conventional optical tweezers. For example, 18 nm diameter gold nanoparticles are trapped in:

Hansen, P. M.; Bhatia, V. K.; Harrit, N.; Oddershede, L. Expanding the Optical Trapping Range of Gold Nanoparticles. *Nano Lett.* 2005, 5, 1937-1942.

and 9.5 nm diameter gold nanoparticles are trapped in:

Hajizadeh, F.; Reihani, S. N. S. Optimized Optical Trapping of Gold Nanoparticles. *Opt. Express* 2010, 18, 551-559.

From my rough calculations, the forces on gold nanoparticles are approximately 10 times stronger than those on polystyrene nanoparticles at this wavelength, but the force scales with particle volume. So the trapping of 9.5 nm diameter gold particles would be approximately as difficult as the trapping of 20 nm diameter polystyrene particles.

If the authors want to claim improved ability to trap the smallest possible particles, then they should quantitatively show how much smaller of a particle they can trap compared to a conventional focused Gaussian beam.

The smallest dielectric particles that were trapped with ACTs were 40 nm in size. To the best of our knowledge, there have been very few experimental attempts to trap such small dielectric nanoparticles with conventional single-beam optical tweezers, since they require very high laser powers. This has been pointed out by multiple authors, including by Ashkin in his pioneering paper (Ref # 1) in 1986. However, trapping of metallic Rayleigh particle is slightly easier due to metal’s higher polarizability than dielectric. In Ref# 17, trapping of 36.5 nm Au particle was performed for the first time by Svoboda and Block where they showed about 7 times higher trapping strength for Au nanoparticle than PS of same size, which was understood by comparing the optical polarizability of Au and PS ($\frac{\alpha_{Au}}{\alpha_{PS}} \sim 7$) at 1047 nm. Later, Hansen *et al* and Hajizadeh *et al* have expanded the trapping range to even smaller metal nanoparticles.

In the table below, we compare the trap stiffness from three highly relevant papers to our measurements. In references 18 and 21, Au particles of size 18 nm and 9.5 nm were used, while in ref. 17, the authors report measurements of 38 nm PS nanoparticle. We have converted the stiffness values to a PS particle of size 40 nm (as in our experiments) using volume scaling of the trapping potential and $\frac{\alpha_{Au}}{\alpha_{PS}} \sim 7$. We find the trap stiffness due to ACT to be significantly higher than traps shown in ref. 17,18 and 21. We have cited all the three papers in appropriate places, and only show the comparison with the traps for dielectric particles in Table 1.

Reference	Au NP size (nm)	Au-Stiffness (κ_p^{Au}) (fNnm ⁻¹ mW ⁻¹)	PS-Stiffness (κ_p^{PS}) ($\kappa_p^{PS} = \kappa_p^{Au}/7$) (fNnm ⁻¹ mW ⁻¹)	Scaled stiffness ($\kappa_p^{PS=40nm}$) (fNnm ⁻¹ mW ⁻¹)
Svoboda et al	36.5	5.0×10^{-3}	0.7×10^{-3}	0.8×10^{-3}

Hansen et al	18	10^{-3}	1.42×10^{-4}	1.6×10^{-3}
Hajizadeh et al	9.5	0.73×10^{-3}	10^{-4}	7.46×10^{-3}
ACT	40 nm PS	-	-	0.245

(2) With regard to minimum irradiance needed on a particle to trap it, there is again no quantitative comparison to a standard Gaussian beam trap. On lines 199-200, the authors state that the irradiance due to the using the ACT is about 10 times lower due to the ACT, making the irradiance at the particle on the order of 0.09 mW/ μm^2 . However it is not clear if this only takes into account the off-axis position of the particle relative to the incident beam, or whether it also takes into account the $\sim 10\text{X}$ plasmonic increase in electric field intensity due to the near-fields of the ACT as shown in Fig. 4c. Also, how do these values compare to what is required using a conventional Gaussian beam trap?

The laser intensity used in our system comparable to the state-of-the-art plasmonic tweezers which is at least 20 times smaller than the intensity typically used with single beam optical tweezers. This is without considering the effective reduction of intensity due to off-peak position of the trapped particle. We have mentioned this clearly in the revised text:

“Please note that, the measurements provided in Fig. 5a correspond to applied laser intensities, and not effective (reduced) intensities, as discussed above.”

The local field enhancement will certainly increase the effective optical incident power; however, the field also decays very quickly away from the ACT. As a result, the enhanced optical field only penetrates a small fraction of the trapped particle, e.g. $\left|\frac{E}{E_0}\right|^2 \sim 10$ only up to about 20 nm from the nanoantenna surface. We have not considered this in our calculations.

(3) Perhaps a deeper potential well depth for nanoparticles could be achieved using the ACT without material damage than is possible with a standard Gaussian beam trap? A recent reference (from our own group) that investigates the maximum potential well depth / maximum force for conventional traps of nanoparticles is:

Melzer, J. E.; McLeod, E. Fundamental Limits of Optical Tweezer Nanoparticle Manipulation Speeds. *ACS Nano* 2018, 12, 2440-2447.

Yes, this is indeed the case for ACT. Due to plasmonic near-field enhancements ACTs can achieve deeper potential well depth as compared to standard Gaussian beam traps, where the depth of the well is defined by the region of highest optical intensity. For example, as per calculations provided in Ref# 5, input power of 100 mW is required to have a well depth of $10k_B T$ for PS particles of size 100 nm. In comparison, as shown in Fig.5b, the same particles will experience a well depth greater than $10k_B T$ under input optical power of 10 mW.

Some other more minor issues that should be addressed prior to publication are:

(a) In lines 13, 128, and 136, I would recommend using the term "ionic solution" rather than "ionic fluid". The terms "ionic fluid" and "ionic liquid" typically refer to substances that are purely composed of ions and are not water-based.

“Ionic fluid” has been changed to “Ionic solution”.

(b) More detail should be given on how the Brownian diffusivities are calculated in Fig. 2c. Currently only a reference to a textbook is given. Do the calculations assume 2D or 3D diffusion? Is the rod modeled as a spherical particle, or is it truly modeled as a rod? If the particle is spherical, what is the equivalent diameter based on: is it particle volume, particle length, or particle width? If the rod is truly modeled as a rod, how is the diffusivity calculated, and is the potential for tumbling taken into account?

We consider diffusion of rod-shaped objects in an unbound (bulk) fluid in three dimensions. This is a reasonable approximation considering the nearest chamber surfaces are more than a micron (order of length of rods) away. The diffusivities provided in the paper correspond to translational diffusion parallel and perpendicular to the major axis of the rod. The formulas used are available in standard literature, including Ref# 22, which depend on the length and diameter of the rod, as well as the viscosity of the surrounding medium. It is possible to use slightly more accurate formulas to estimate the diffusivities (e.g. Carrasco, Beatriz, and José García de la Torre. "Hydrodynamic properties of rigid particles: comparison of different modeling and computational procedures." *Biophysical journal* 76, no. 6 (1999): 3044-3057.), however, this does not change the key message: 3- μm rods are significantly less diffusive than 50 nm discs. The rotational diffusivities (related to tumbling as mentioned by the reviewer) can also be considered in the same way; however, the conclusions remain unchanged and therefore not discussed.

(c) In the supplemental info and in the Fig. 4 caption, the plasmonic tools appear to be called CPTs rather than ACTs. The acronym should be consistent.

All the acronyms have been changed to ACT.

(d) On line 152, the authors write that the laser spot size was increased to 10 μm "through further defocusing". It is not clear to me what is happening here. I would have thought that defocusing of the objective would primarily move the trap along the z-axis, but not significantly change the spot size. Is this a 2D trap and not a 3D trap? Or is it a different lens in the optical system that is being defocused?

The defocusing of the of the laser beam move the trap along beam axis. By tuning the beam divergence in the objective entrance aperture, we position the laser focus at a location above the fluid chamber which is inaccessible to the colloids. Now as one moves along beam axis (say, along z direction) away from the laser focus the beam width increases and appears to be a larger spot size at the plane of observation (x-y plane) inside the fluidic volume. This forms a weak quasi-two-dimensional trap which loosely holds the ACTs and colloids. The following has been added in main text.

“When the laser spot size was increased ($\sim 10 \mu\text{m}$) through further defocussing, a wide shallow trap was formed, which is referred to as “optical bowl” in the literature³⁰⁻³². This was achieved by focusing the laser beam at a plane higher (along beam propagation direction, i.e. z-axis) than the plane of observation inside a microfluidic chamber, resulting in a weak two-dimensional trap. The beads experience radiation pressure upward which is balanced by the electrostatic repulsion due to the top surface of the fluidic chamber. The z-position of the focal plane was controlled by tuning the divergence of the beam entering the microscope objective. Here the ACT achieved a horizontal orientation (see right panel, Fig. 4a) loosely trapped within the illuminated region.”

(e) On line 156, the authors write, "the illumination wavelength was found to be slightly red-shifted to the resonant modes." I do not understand this. I thought the illumination wavelength was fixed at 1064 nm. Is there some nonlinear process that is changing the illumination wavelength inside the system?

We have re-worded this sentence. The illumination wavelength is fixed at 1064 nm which is slightly detuned (red-shifted) from the plasmon resonance wavelength (~ 1000 nm). The sentence has been modified accordingly.

“In Fig. 4b, we show the calculated extinction spectrum of the nanodisks as a function of wavelength, where the illumination wavelength was chosen to be slightly detuned (red-shifted) to the resonant modes.”

(f) Where were all of the different nanoparticles acquired? Were they bought from a company or synthesized?

All the nanoparticles were purchased from these following companies.

0.4, 0.7, 1.0 μm PS: Spherotech INC.

40 and 200 nm PS: Bangs Labs INC.

100 fluorescent ND: FND Biotech INC.

300 nm magnetic particle: Spherotech INC.

This information has been added in respective places of the manuscript.

Sincerely,
Euan McLeod
College of Optical Sciences
University of Arizona

Reviewer #2 (Remarks to the Author):

Dear Editor,

This paper reports the dynamic manipulation of particles by employing an ACTS. The authors successfully fabricated metallic nanodisks of varying diameter and fixed thickness on dielectric nanorod. They, further, trapped 100 nm fluorescent nanodiamonds with intensity $4 \text{ mW}/\mu\text{m}^2$, beads in PBS solution. However, major part of results in this paper were done by COMSOL, and experimental part was very poor and lack of experimental analysis, such as trapping force, stiffness, quality factor and dynamic motions of the trapping system of ACT itself, ACT with NV diamonds and ACT with 200 nm polystyrene.

Based on suggestions made by the reviewers, we have made many changes in the manuscript which have significantly improved the manuscript. In the present version we have added experimental measurement of trap stiffness of ACT and compared it with other kinds of plasmonic traps and standard Gaussian beam traps in Table 1. We have achieved about 200 times higher trap stiffness for a 40 nm particle than conventional optical traps which is also comparable to other kinds of plasmonic traps. From Fig. 5c we have also measured the stiffness of the optical trap that holds the ACT. In addition, we have experimentally measured the temperature increase of the ACTs which matched well with the simulated results. The present version of the manuscript has a good balance between experimental and numerical results, which we hope will answer the criticisms by the reviewer.

All these ACT, NV diamond, and 200 nm polystyrene can be easily trapped separately at different and much lower intensity levels with other systems. Their trapping of group particle seems to be not so much different in term of physics with a group of particles of clusters from Toussaint and Bryan in a Nano Letter sometimes back. I am not sure this is enough to justify publication in Nan. Com; especially considering there is a priori nothing novel about plasmon-assisted trapping and this configuration does not perform particularly better than prior ones. The reviewer feels that the scope of this manuscript may be more suited to another journal after lots of corrections of course, and for sure it is not for a Nature Com.

We respectfully disagree with the reviewer on the first point, which could have happened due to a misinterpretation of our experimental strategy. We have introduced and trapped ACTs for the first time in this manuscript. Trapping NV diamonds and 200 nm PS particles by plasmonic tweezers have been done using comparable optical intensities (see Table 1 of the main text for a detailed comparison of stiffnesses of different traps, including ACTs) as used in other plasmonic tweezers. Crucially, these techniques using plasmonic tweezers do not allow maneuverability possible with ACTs and herein lies one of the main novelties of this study: ACTs can overcome limitations with existing tweezer technologies.

The same argument can be made regarding the paper (Ref# 43) suggested by the reviewer pertaining to trapping of multiple particles, which occurs on a nanopatterned plasmonic substrate. Unlike this prior work, as shown in movie M4, we demonstrate trapping of multiple particles using multiple ACTs in a loosely focused optical trap without the requirement of specialised nanopatterned plasmonic substrate.

To highlight more on the novelty issue, the main idea of the present research is to combine the strengths of two powerful manipulation techniques: optical and plasmonic tweezers. This has been a standing goal of the community, articulated in a recent review article (Ref# 15) as "*If a technique is*

developed that provides us with the ability to work in the nanoregime with the same degree of control as conventional optical tweezers provide in the microregime this will prove to be a huge boon to the scientific community at large”.

Our approach is based on a “tweezer within a tweezer” concept, where we use metal-dielectric hybrid nanostructures that can be trapped and maneuvered by conventional optical tweezers and the trapped ACTs can simultaneously generate strongly confined optical near fields in their vicinity, functioning as near-field plasmonic traps themselves for colloids as small as 40 nm. These “Active Colloidal Tweezers” (or ACTs) can overcome certain long-standing limitations in current state-of-the-art optical manipulation:

- (i) Their trapping performance is not limited by the diffraction limited focusing of light, **allowing trapping of smaller colloids at lower laser powers** as in plasmonic tweezers.
- (ii) They can be **actively maneuvered** to allow selective and parallel cargo transport as in conventional optical tweezers.
- (iii) ACTs can be operated in standard microfluidic chambers, implying specialized nanophotonic substrates are not necessary, as in conventional optical tweezers.
- (iv) Due to its all-optical nature, ACTs are applicable in different classes of media, including ionic solutions.

The trapping demonstrations clearly shows the great promise of this technique. Of special importance is the active manipulation of very small (40 nm particles) colloids which is impossible to achieve in conventional optical traps (using typical laser powers). We have also shown manipulation of 200 nm particles in bio-fluids where many of the current techniques fail due to the inherent ionic nature of biological media.

Manipulation aspects	Optical tweezer	Plasmonic tweezer	Our work (submitted)
Nanoscale trapping	No	Yes	Yes
Powerful lasers	Yes	No	No
Trapping type	Dynamic	Static	Dynamic
Substrate	Any	Metallic, nanopatterned	Any
Transport	Yes	No	Yes

Some comments:

1) Figure 5 c, I don't know what they want to show in this figure, it looks not like a near-field, the laser beam was not scattered and so on,.. with the presence of object. It is meaning less that figure. Again similar problems with the figure 6a and it is hard to convince people that this is a trap for few seconds or it was just the flow of water that caused by the heating and this may lead the particle flowing around the ACT and got stuck for few second and release.

We understand the possible confusion over the previous representation of Fig. 5c. Therefore, we have made a new schematic for clearer representation (now Fig. 5e) and the following has been added in main text.

“In Fig. 5e, we highlight the position of the trapped beads within the focal volume. Without the ACT, a particle can be trapped at a laser focus where the trapped object is located at the region of highest intensity. However, in the present scheme, ACTs occupy the position of maximum intensity implying the effective optical power incident on the trapped colloids is a fraction of applied power. For a 400 nm PS bead, this correspond to a further reduction of effective incident laser power by a factor of ~ 10 , which can provide crucial advantages in biomanipulation applications³⁹.”

Regarding Fig 6a, the trapping of one 40 nm particle is clearly shown with red arrow which is trapped for about 25 seconds with the ACT, where the trapping and release could be controlled by external illumination. Regarding flow of water, previous literature has shown (ref: Donner, Jon S., et al. "Plasmon-assisted optofluidics." *Acs nano* 5.7 (2011): 5457-5462.) negligible (~ 1 nm/s) convective velocity of an isolated plasmonic antenna for temperature rise as high as 80 K, so convective flow resulting in hydrodynamic trapping of the 40 nm particle can be ruled out.

2) There is also a lack of discussion of where the trapped particle is in three dimensions, which is important for a broadly read journal such as *Nat. Com.*, rather than those who are experts in nanoplasmic structures.

The ACTs are held at two different configurations as shown in Fig. 4. At vertical configuration the ACT is held at an optical tweezer focus, whereas a horizontal configuration is achieved with a loosely focused two-dimensional trap.

“The optical property of an ACT was dominated by the highly polarizable silver nanodisk, which was trapped at the focal spot. For tight focussing (spot size ~ 1 μm), the ACT was oriented along the beam propagation direction (i.e. z-axis in Fig. 4a), with the dielectric rod always remaining under the focus. This occurred because the silver nanodisk was subject to higher radiation pressure compared to the silica part, which pushed the metal part higher as the ACT got trapped in the vertical orientation. When the laser spot size was increased (~ 10 μm) through further defocussing, a wide shallow trap was formed, which is referred to as “optical bowl” in the literature³⁰⁻³². This was achieved by focusing the laser beam at a plane higher (along beam propagation direction, i.e. z-axis) than the plane of observation inside a microfluidic chamber, resulting in a weak two-dimensional trap. The beads experience radiation pressure upward which is balanced by the electrostatic repulsion due to the top surface of the fluidic chamber. The z-position of the focal plane was controlled by tuning the divergence of the beam entering the microscope objective. Here the ACT achieved a horizontal orientation (see right panel, Fig. 4a) loosely trapped within the illuminated region.”

Reviewer #3 (Remarks to the Author):

In the manuscript: "All optical dynamic nanomanipulation with active colloidal tweezers," the authors present a hierarchical scheme for manipulating small objects with plasmonic tweezers attached to nanorods. This is an original approach to achieve such manipulation that will be of great interest to the community and I believe that it may be suitable for Nature Communications. I do have some comments that should be addressed prior to publication.

We thank the reviewer for her/his constructive comments and suggestions. We have incorporated the suggested changes in the revised version, and this has resulted in a significantly improved manuscript.

1) My greatest concern is that the inclusion of an isolated plasmonic structure will result in substantial heating at even moderate powers. This is basically the premise of thermoplasmonics. The authors say that the heating was minimal and they provide a plot to show only a few degrees temperature increase; whereas other works have shown hundreds of degrees temperature increase. Is there any experimental evidence (e.g. changes in fluorescence, changes in Brownian motion of trapped particle) to support their claim?

This point raised by the reviewer has been considered in detail in the revised manuscript. The choice of silver as the plasmonic material, dimensions of the plasmonic disc and using 1064 nm as the illumination wavelength; all of these ensured the temperature rise to be smaller than most thermoplasmonic systems that are typically made with gold. To check the numerical predictions, we have performed new experiments to measure the temperature rise using changes in fluorescence of the dye molecules in the heated region (as suggested by the reviewer). The experimental results were found to agree with the numerical estimates. These have been included in section VII-VIII of the revised SI, and the following has been added to the main text.

“Also, relevant is the temperature rise in the system due to absorption of incident light. Using temperature dependence of fluorescence of the dye molecules in the trapping region, we have measured (see supporting information, section VIII) the temperature rise to be $\sim 13\text{ K}$ at an intensity level $10\text{ mW}/\mu\text{m}^2$, which corresponds to a trapping potential larger than $10\text{ }k_B T$ for a 100 nm particle. Note this is lower than the optical damage threshold of a living bacteria ($\sim 18\text{ mW}/\mu\text{m}^2$)³⁹ implying interesting possibilities in biomanipulation experiments.”

2) The authors are quite careful to say that the present scheme allows for lower illumination intensities (note typo "illuminations intensities"). They are consistent in this claim saying that the overall power is lower to trap. This is true and certainly a good feature; however a common error in plasmonic trapping is to say that the plasmonic particle creates greater forces that are required to overcome diffusion of smaller particles. The authors make similar claims in the same sentence referring to the larger gradient forces. "These near-field based tweezers can generate stronger gradient forces and thereby able to trap smaller colloids". It is not the strength of the gradient force that allows for trapping smaller particles -- it is the local intensity. The potential well created by that intensity has a depth defined by the highest intensity region. It does not matter how localized or spread out that intensity is -- the depth of the well is defined by that intensity, and so the greater forces by plasmonic tweezers do not allow trapping smaller particles. They give greater

localization of the trap, and trapping with lower overall power, but a conventional tweezer system will trap particles of the same size as a plasmonic one if the local intensity at the maximum is the same. Making this point clear in the manuscript is suggested, since it is misleading at best (and incorrect at worst) the way it is presented. (Again later in the manuscript they say "... sufficient to generate enough gradient force to trap..." -- also giving the wrong impression that greater gradient forces allow trapping smaller particles, when energy-based considerations show that smaller forces over a larger distance would be just as good. A general readthrough of the manuscript to carefully eliminate these types of claims is suggested.)

We agree with this criticism and this has been incorporated in the revised manuscript in all the appropriate places. In addition, we have changed the quantity plotted in Fig. 5b from force to potential representation.

REVIEWERS' COMMENTS:

Reviewer #1 (Remarks to the Author):

In my opinion, the authors have sufficiently addressed all of the points raised by the reviewers in their revised manuscript.

Reviewer #2 (Remarks to the Author):

Dear Editor,

The manuscript has been improved by the authors after the reviewers comments. However, the reviewer feels that it still not suitable for the Nature Communications. The reviewers feeling is that the manuscript entitled "All optical dynamic nanomanipulation with active colloidal tweezers" provides a novel approach of trapping technique with promising future applications and this could be suitable for Scientific Reports or similar journal. The reviewer has added some comments that support his/her opinion.

The trapping stiffness is still lower than the corresponding one from SIBA trapping (Table 1). The improvement of 200 times higher than standard Gaussian beam trap is good but in literature there is plasmonic tweezers design which provides 1000 enhancement compared with standard tweezers (Xue et al Photonics Research).

A paper of Gordon et al (Low-Power Optical Trapping of Nanoparticles and Proteins with Resonant Coaxial Nanoaperture Using 10 nm Gap) demonstrated trapping of protein and polystyrene beads diluted in buffer solution (ionic solution) with low incident laser power i.e. 4.5 mW. Additionally, Gordon noted that "One main advantage of using coaxial apertures or other aperture-based trapping devices is the rapid heat dissipation upon illumination".

Moreover, Recce et al (Manipulating the Quantum Coherence of Optically Trapped Nanodiamonds) optically trapped nanodiamonds and demonstrated spin relaxometry measurements. The authors only trapped and released nanodiamonds using ACTs configuration which it is very important but it is something that reported already in literature.

The authors trapped particle with 40 nm diameter. However, in literature many polystyrene particles with smaller diameter than 40 nm have been reported using plasmonic tweezers with lower incident intensities (see Gordon et al and Quidant et al works).

The transport of nanoparticle has been demonstrated in "Nano-optical conveyor belt, part II: demonstration of handoff between near-field optical traps" using -shaped structure.

The maneuverability is very important for tiny particle likes proteins or virus which their dimensions are lower than 40 nm. The authors proved the maneuverability for "big" particles compared with bio-specimens.

Thank you

Reviewer #3 (Remarks to the Author):

The authors have suitably addressed my concerns.

Reviewer #1 (Remarks to the Author):

In my opinion, the authors have sufficiently addressed all of the points raised by the reviewers in their revised manuscript.

We thank the reviewer for carefully reviewing the work and for his constructive comments which have significantly improved the work.

Reviewer #2 (Remarks to the Author):

Dear Editor,

The manuscript has been improved by the authors after the reviewers comments. However, the reviewer feels that it still not suitable for the Nature Communications. The reviewers feeling is that the manuscript entitled "All optical dynamic nanomanipulation with active colloidal tweezers" provides a novel approach of trapping technique with promising future applications and this could be suitable for Scientific Reports or similar journal. The reviewer has added some comments that support his/her opinion.

The trapping stiffness is still lower than the corresponding one from SIBA trapping (Table 1). The improvement of 200 times higher than standard Gaussian beam trap is good but in literature there is plasmonic tweezers design which provides 1000 enhancement compared with standard tweezers (Xue et al Photonics Research).

The SIBA mechanism is different from conventional plasmonic trapping schemes and has the highest trap stiffness. Crucially, SIBA based techniques have been demonstrated in devices where the nanoantennas are tethered to a surface, e.g. the work by Xue et al. This is different from our work, which was not aimed to achieve the highest trapping stiffness, but rather demonstrate dynamic capabilities for optical nanomanipulation. In future, it may be possible to integrate SIBA-mechanism with the design of ACT to improve the trapping stiffness.

A paper of Gordon et al (Low-Power Optical Trapping of Nanoparticles and Proteins with Resonant Coaxial Nanoaperture Using 10 nm Gap) demonstrated trapping of protein and polystyrene beads diluted in buffer solution (ionic solution) with low incident laser power i.e. 4.5 mW. Additionally, Gordon noted that "One main advantage of using coaxial apertures or other aperture-based trapping devices is the rapid heat dissipation upon illumination".

Gordon *et al* showed improved trapping performance in ionic fluid but with static plasmonic traps (Ref #11) unlike our work, which is about dynamic nanomanipulation. Regarding the heating, we have already discussed this in detail in Supplementary Note 2 and 3.

Moreover, Recce et al (Manipulating the Quantum Coherence of Optically Trapped Nanodiamonds) optically trapped nanodiamonds and demonstrated spin relaxometry measurements. The authors only trapped and released nanodiamonds using ACTs configuration which it is very important but it is something that reported already in literature.

The trapping and release of nanodiamonds shown in our paper is an example of dynamic manipulation of small colloids at low optical intensities, which is different from what has been

reported before. However, the paper suggested by the reviewer is relevant and therefore cited (ref# 28) in the revised manuscript.

The authors trapped particle with 40 nm diameter. However, in literature many polystyrene particles with smaller diameter than 40 nm have been reported using plasmonic tweezers with lower incident intensities (see Gordon et al and Quidant et al works).

The transport of nanoparticle has been demonstrated in “Nano-optical conveyor belt, part II: demonstration of handoff between near-field optical traps” using –shaped structure. The maneuverability is very important for tiny particle likes proteins or virus which their dimensions are lower than 40 nm. The authors proved the maneuverability for “big” particles compared with bio-specimens.

Thank you

Previous literature has showed trapped of smaller particles but with a plasmonic trap fixed on substrate. In contrast, our work is focused on dynamic nanomanipulation. Among reviewer’s suggestions, we have cited the paper by Zheng et al. on “Nano-optical conveyor belt” in Ref# 15, which, please note, requires a nanopatterned surface and has a small working range of 4.5 μm .

Also note, 40 nm is not the limit of our technique. It may be possible to extend ACT capabilities to even smaller colloids which we have not tried yet.

Reviewer #3 (Remarks to the Author):

The authors have suitably addressed my concerns.

We thank the reviewer for carefully reviewing the work and for his constructive suggestions which have significantly improved the work.